**Investigation**

# Breakpoint–chiasma interference in pericentric inversion heterokaryotypes

Øystein Kapperud (ID) *

Norwegian Institute of Public Health, IT Systems Oslo, P.O. Box 222 Skøyen, Oslo 0213, Norway

*Corresponding author: Norwegian Institute of Public Health, IT Systems Oslo, P.O. Box 222 Skøyen, Oslo 0213, Norway. Email: oystein.kapperud@fhi.no

Heterozygous inversion breakpoints inhibit the formation of chiasmata in their vicinity, and it has been suggested that they do so through the same mechanism that also causes interference between chiasmata. In this paper, I therefore extend my earlier model of chiasma interference to account for interference between breakpoints and chiasmata in pericentric inversion heterokaryotypes. Using this model to analyze recombination and sterility datasets for *Drosophila melanogaster*, I find support for the hypothesis that inversion breakpoints interfere with chiasmata in the same way and to the same degree that other chiasmata do. I also find that breakpoints, like chiasmata, appear to show negative interference in the pericentromeric region, and positive interference elsewhere. I discuss the implications of these findings in light of the recent HEI10 coarsening interference hypothesis, and conclude with some remarks about the evolutionary origin of chiasma interference.

Keywords: chiasma interference; meiosis; chromosomal inversions

## Introduction

It is well established that chiasma formation is partly suppressed inside and immediately outside of inversions in heterokaryotypes (*e.g.* Sturtevant and Beadle 1936; Novitski and Braver 1954; Coyne *et al.* 1993; Navarro and Ruiz 1997; Pegueroles *et al.* 2010), but the cause of this effect is disputed. The long prevailing theory that the chiasma suppression is caused by imperfect synapsis is contradicted by the finding that it also occurs in heterokaryotypes that show close to normal levels of pairing, synapsis, and double-strand-breaks (Gong *et al.* 2005). An alternative theory, suggested by Gong *et al.* (2005), is that structural similarities between heterozygous inversion breakpoints and chiasmata cause the breakpoints to interfere with the generation of nearby chiasmata in the same way that other chiasmata do. The theory therefore holds that *breakpoint–chiasma interference* is a special case of the well-known phenomenon of *chiasma interference* (*e.g.* Muller 1916; Foss *et al.* 1993; McPeek and Speed 1995; Zhao *et al.* 1995; Copenhaver *et al.* 2002; Kapperud 2023). As Gong *et al.* (2005) noted, this can explain why chiasma formation is suppressed despite near-perfect synapsis, why the suppression primarily occurs close to breakpoints, and why it is much less pronounced in a species with weaker chiasma interference (Sherman and Helms 1978).

Recently, Koury (2023) emphasized that the theory also explains the finding that chiasma formation is suppressed *outside* of, as well as inside of, inversions in heterokaryotypes. This suppression, he argued, typically extends too far outside of the inversion to be explainable by imperfect synapsis, but it is easily explained if the hypothesized interference occurs in both directions from breakpoint sites. To test this idea, he experimentally determined the recombination rates of intervals at different positions outside of a paracentric inversion in *Drosophila*, and compared the results to those expected

from three different models: one in which breakpoints do not cause interference, one in which breakpoints do cause interference, and one in which breakpoints cause interference, but not across centromeres, in accordance with the finding that the centromere appears to block the interference from chiasmata in *Drosophila* (Graubard 1932; Stevens 1936; Kapperud 2023). As expected, the third model proved superior in predicting the observed data.

*Counting models* of chiasma interference model the genetic distance between chiasmata as a (weighted) sum of the distances between a given number (or numbers) of independently distributed dummy events (Cobbs 1978; Stam 1979; Foss *et al.* 1993; Foss and Stahl 1995; Zhao *et al.* 1995; Lange *et al.* 1997; Navarro *et al.* 1997; Copenhaver *et al.* 2002; Nolan 2017; Kapperud 2023). Such models have the advantage of enabling analysis of recombination data for multiple intervals even when the position of the chiasmata within each interval is unknown. In a recent paper (Kapperud 2023), I presented a general counting model of chiasma interference in inversion homo- and heterokaryotypes, and suggested that the interference either crosses or is blocked by the breakpoint boundaries. Unbeknownst to me at the time, the breakpoint–chiasma interference hypothesis suggests a third possibility. Hence, in this paper, I will show how to calculate recombination pattern proportions for an indefinite number of loci positioned inside and outside of a pericentric inversion, with and without breakpoint–chiasma interference. I will then use this model to perform maximum likelihood parameter estimations on *Drosophila* recombination data (from Roberts 1967), so as to determine if incorporating breakpoint–chiasma interference improves the fit. I will also show how to incorporate breakpoint–chiasma interference into the sterility equations presented in my previous paper, and fit this extended model to *Drosophila* sterility data from Coyne *et al.* (1993).

## Models

Koury (2023) noted that the challenge in modeling breakpoint–chiasma interference *inside* of inversions is to determine how the interferences from the two breakpoints interact in the middle. Luckily, this is not a problem in *Drosophila* pericentric inversion heterokaryotypes, since the interference appears to be blocked by the centromere in this genus (Graubard 1932; Stevens 1936; Kapperud 2023; Koury 2023). Hence, recombination in such individuals can be modeled by dividing the chromosome into the four regions separated by the breakpoints and the centromere, computing recombination pattern probabilities for each region individually, and then combining the results (see Fig. 1). For simplicity, I will assume that the chiasmata are all generated by a single pathway (see Kapperud 2023 for an extended model with two pathways). The genetic distances between the chiasmata are modeled as a weighted sum of exponential distributions. This is achieved by assuming that each pair of chiasma events is separated by a number of independently distributed dummy events, and that the probability of observing $i$ intervening dummy events is given by $\gamma_i$, so that $\sum_{i}^{m} \gamma_i = 1$. $m$ is the upper limit of non-zero values of $\gamma_i$, so that $\gamma_i = 0$ for all $i > m$ (see Table 1 for an overview of the notation used in this text). If we define the *phase* at a given location on the chromosome as the number of dummy events between that location and the nearest chiasma event in a given direction, then the recombination pattern probabilities can be found by summing over all possible phases at each marker loci by means of matrix multiplication (Zhao *et al.* 1995). For a region with $n$ intervals, if $r_k$ signifies recombination ($r_k = 1$) or non-recombination ($r_k = 0$) in interval $k$ for a given recombination pattern $\mathbf{r} = \{r_0, r_1, r_2, \ldots, r_{n-1}\}$, then the probability of observing $\mathbf{r}$ is given by

$$\Pr(\mathbf{r}) = \mathbf{\Omega} \left( \prod_{k=0}^{n-1} \mathbf{M}_k(\mathbf{r}) \right) \mathbf{1}^T \tag{1}$$

where $\mathbf{\Omega} = \{\Omega_0, \Omega_1, \Omega_2, \ldots, \Omega_m\}$ is a vector that gives the phase distribution at the starting point, $\prod_{k=0}^{n-1} \mathbf{M}_k(\mathbf{r}) = \mathbf{M}_0(\mathbf{r})\mathbf{M}_1(\mathbf{r}) \cdots \mathbf{M}_{n-1}(\mathbf{r})$ and matrix $\mathbf{M}_k(\mathbf{r})$ is given by

$$\mathbf{M}_k(\mathbf{r}) = \begin{cases} \frac{1}{2}\mathbf{G}_k & \text{if } r_k = 1 \\ \frac{1}{2}\mathbf{G}_k + \mathbf{H}_k & \text{if } r_k = 0 \end{cases} \tag{2}$$

where element $i, j$ of matrices $\mathbf{H}_k$ and $\mathbf{G}_k$ are respectively given by

$$\mathbf{H}_k[i, j] = \begin{cases} \frac{\lambda_k^{i-j} e^{-\lambda_k}}{(i-j)!} & \text{if } i \geq j \\ 0 & \text{if } i < j \end{cases} \tag{3}$$

and

$$\mathbf{G}_k[i, j] = \sum_{t=0}^{\infty} b_t \sum_{s=j}^{m} \gamma_s \frac{\lambda_k^{i+1+t+s-j} e^{-\lambda_k}}{(i+1+t+s-j)!} \tag{4}$$

where

$$b_t = \begin{cases} \sum_{a=0}^{t-1} b_a \gamma_{t-1-a} & \text{if } t = 1, 2, 3 \cdots \\ 1 & \text{if } t = 0 \end{cases} \tag{5}$$

and

$$\lambda_k = 2l_k' \sum_{s=0}^{m} (s+1)\gamma_s \tag{6}$$

where $l_k'$ is the genetic length in Morgans of interval $k$ for the heterokaryotype, and $\lambda_k$ is the expected number of events (dummy events + chiasma events) in interval $k$ for the heterokaryotype (see Lange *et al.* 1997 or Kapperud 2023 for a derivation). If the genetic length of an interval $k$ in a homokaryotpe is known and denoted $l_k$ then $d_k = \frac{l_k'}{l_k}$ indicates the degree to which the presence of the inversion alters the genetic map. Note that the recombination patterns are calculated linearly in a given direction purely as a matter of mathematical convenience; the model does not necessarily assume that the interference actually propagates in this way.

Of particular interest for the present discussion are the parameters $\mathbf{\Omega}$ and $d_k$. In Kapperud (2023), $\mathbf{\Omega}$ is assumed to always be equal to the stationary phase distribution $\boldsymbol{\pi}$ for which element $i$ is given by

$$\pi_i = \frac{\sum_{s=i}^{m} \gamma_s}{\sum_{s=0}^{m} (s+1)\gamma_s} \tag{7}$$

In this case, there is no interference between the starting point of the investigation and nearby chiasmata. The breakpoint–chiasma interference hypothesis simply states that $\mathbf{\Omega} = \boldsymbol{\gamma} = \{\gamma_0, \gamma_1, \gamma_2, \ldots, \gamma_m\}$ when the four regions are analyzed starting from the adjoining breakpoint (see Fig. 1). In this case, the interference from the breakpoint is equal to the interference caused by a chiasma event. $\mathbf{\Omega}$ can also be set to other values, so as to model the hypothesis that breakpoint–chiasma and chiasma–chiasma interference differ in degree, as discussed further below. The parameters $d_k$ are of interest because they can be interpreted as the degree to which chiasma suppression in heterokaryotypes is caused by other phenomena than interference from the breakpoints. That is, if *all* of the chiasma suppression is caused by breakpoint interference, then $\mathbf{\Omega} = \boldsymbol{\gamma}$ and $d_k = 1$ for all $k$. If, on the other hand, $\mathbf{\Omega} = \boldsymbol{\gamma}$ and $d_k < 1$ for some $k$, then some of the suppression is caused by imperfect synapsis or other factors.

Recombination inside the inverted region creates *unbalanced* gametes that do not have a full set of genes, and are therefore inviable (*e.g.* Sturtevant and Beadle 1936; Coyne *et al.* 1993; Navarro and Ruiz 1997; Navarro *et al.* 1997; Kapperud 2023). In pericentric inversion heterokaryotypes, a gamete will be unbalanced if and only if it shows recombination in *either* region 1 *or* region 2 (but not both) in Fig. 1 (see Kapperud 2023). The proportion of such inviable gametes is called the sterility. Like for the recombination pattern expression above, I find the relevant expression for the sterility ($\zeta$) by replacing the initial phase distribution with $\mathbf{\Omega}$ in the expressions derived in Kapperud (2023). For a pericentric inversion with total heterokaryotype genetic length $l'$ (in Morgans) and two inner intervals (enclosed by the centromere and one of the breakpoints; see regions 1 and 2 in Fig. 1) of respective genetic lengths $l'\rho$ and $l'(1-\rho)$ (for $0 \leq \rho \leq 1$), this gives

$$\zeta = \Pr(r_1 = 1)\Pr(r_2 = 0) + \Pr(r_1 = 0)\Pr(r_2 = 1) \tag{8}$$

where

$$\Pr(r_k = 1) = \frac{1}{2}\left(1 - \sum_{s=0}^{m} \Omega_s \sum_{c=0}^{s} \frac{e^{-\lambda_k} \lambda_k^c}{c!}\right) \tag{9}$$

$$\lambda_k = 2l_k' \sum_{s=0}^{m} (s+1)\gamma_s \tag{10}$$

$$l_k' = \begin{cases} l'\rho & \text{if } k = 1 \\ l'(1-\rho) & \text{if } k = 2 \end{cases} \tag{11}$$

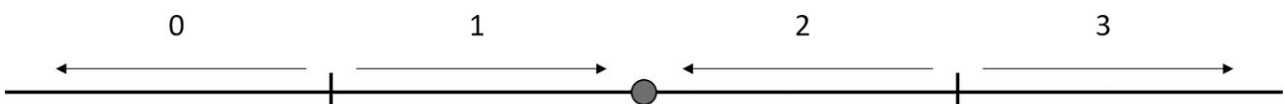

**Fig. 1.** The four independent regions of a chromosome with a pericentric inversion. The circle represents the centromere and the vertical lines represent the inversion breakpoints. The recombination pattern probabilities are calculated in each region independently starting from the adjoining breakpoint, as indicated by the arrows.

**Table 1.** Notation table.

| Symbol | Description | Definition |
|---|---|---|
| $\gamma = \{\gamma_0, \gamma_1, \gamma_2, \ldots, \gamma_m\}$ | A probability distribution for the number of independently distributed dummy events between a pair of chiasma events. | |
| $\Omega = \{\Omega_0, \Omega_1, \Omega_2, \ldots, \Omega_m\}$ | A probability distribution for the number of independently distributed dummy events between an inversion breakpoint and the nearest chiasma event in a given direction. | |
| $\pi = \{\pi_0, \pi_1, \pi_2, \ldots, \pi_m\}$ | The stationary phase distribution. | $\pi_i = \frac{\sum_{s=i}^{m} \gamma_s}{\sum_{s=0}^{m} (s+1)\gamma_s}$ |
| $H_0$ | The hypothesis that the interference signal is blocked by the centromere and the inversion breakpoints, and that neither interferes with the formation of nearby chiasmata. | $\Omega = \pi$ |
| $H_1$ | The hypothesis that the interference signal is blocked by the centromere, and that the inversion breakpoints interfere with nearby chiasmata to the same degree that other chiasmata do. | $\Omega = \gamma$ |
| $H_2$ | The hypothesis that the interference signal is blocked by the centromere, and that the inversion breakpoints interfere with nearby chiasmata, but not necessarily to the same degree that other chiasmata do. | $\Omega \neq \pi \neq \gamma$ |
| $[q]$ | The integer part of a decimal number $q$. | |
| $\{q\}$ | The fractional part of a decimal number $q$. | |
| $q_c$ | The strength of interference between chiasma events. | $\gamma_{[q_c]} = 1 - \{q_c\}, \gamma_{[q_c]+1} = \{q_c\}$ |
| $q_b$ | The strength of interference between inversion breakpoints and chiasma events. | $\Omega_{[q_b]} = 1 - \{q_b\}, \Omega_{[q_b]+1} = \{q_b\}$ |
| $l'_k$ | The genetic length (in Morgans) of interval $k$ in a heterokaryotype. | |
| $l_k$ | The genetic length (in Morgans) of interval $k$ in a homokaryotype. | |
| $d_k$ | The inhibition of chiasma formation that is not due to breakpoint–chiasma interference, in interval $k$ of an inversion heterokaryotype. | $d_k = \frac{l'_k}{l_k}$ |
| $I'$ | The gentic length (in Morgans) of an inversion in a heterokaryotype. | |
| $I$ | The genetic length (in Morgans) of an inversion in a homoaryotype. | |
| $d_I$ | Same as $d_k$, but for the whole inverted region. | $d_I = \frac{I'}{I}$ |
| $\rho$ | The relative position of the centromere in the inverted region. | |

$$\Pr\{r_k = 0\} = 1 - \Pr\{r_k = 1\} \qquad (12)$$

$\rho$ is here the relative position of the centromere inside the inversion. Similarly to the recombination case, if the total genetic length of the inverted region in a homokaryotype is $I$, then $d_I = \frac{I'}{I}$ indicates the degree to which the chiasma suppression is caused by factors other than interference.

For simplicity and parameter interpretability, I will assume the *gamma model* parametrization in Kapperud (2023), whereby the strength of chiasma interference is given by a single non-integer parameter $q_c$ so that $\gamma_{[q_c]} = 1 - \{q_c\}, \gamma_{[q_c]+1} = \{q_c\}$, where $[q_c]$ and $\{q_c\}$ are the integer part and fractional part, respectively, of $q_c$ (this is equivalent to the *mixture* model of Lange et al. (1997), although they use two parameters rather than one). I will consider three hypotheses. The first, $H_0$, assumes that the chiasma interference is blocked by the breakpoints, and that the breakpoints do not interfere with chiasma formation. The second model, $H_1$, assumes that breakpoints interfere with chiasma formation in the same way and to the same degree that other chiasmata do. This is equivalent to model $H_2$ in Koury (2023). The third model, $H_2$, assumes that breakpoints interfere with chiasma formation, but it allows the strengths of breakpoint–chiasma and chiasma–chiasma interference to differ. If the strength of breakpoint interference is $q_b$, then the phase distributions at the breakpoints is here given by $\Omega_{[q_b]} = 1 - \{q_b\}, \Omega_{[q_b]+1} = \{q_b\}$. The three hypotheses can hence be stated concisely as $H_0 : \Omega = \pi$, $H_1 : \Omega = \gamma$, $H_2 : \Omega \neq \pi \neq \gamma$. All these

hypotheses assume that the interference is blocked by the centromere, in accordance with Koury's (2023) results. Koury (2023) also discussed the possibility that the presence of an inversion breakpoint in the vicinity of the centromere will cause increased rather than decreased recombination due to negative breakpoint–chiasma interference in this region. Negative chiasma interference, meaning that chiasmata encourages rather that discourages the formation of other nearby chiasmata, has been demonstrated in the pericentromeric region of *Drosophila* (Green 1975; Sinclair 1975; Denell and Keppy 1979). Therefore, if breakpoint interference is caused by the same mechanism, we should expect that the same applies to interference between breakpoints and chiasmata. This negative interference is region-specific and *not* blocked by the centromere and therefore difficult to model directly, but it can be detected as an increase in the estimated genetic lengths ($d_k > 1$) of intervals that contain a breakpoint close to the centromere.

To illustrate the effect of interference on recombination in pericentric inversion heterokaryotypes, consider a case of three loci, A, B and C, where A is perfectly linked to the left breakpoint (so that there is no recombination between these), B is perfectly linked to the centromere, and C is perfectly linked to the right breakpoint. The only way for these three loci to be broken up through recombination is if there is recombination between A and B *and* recombination between B and C, since recombination in only one of these intervals will result in inviable gametes. This is equivalent to observing recombination in regions 1 and 2 in Fig. 1. Figures 2 and 3 show that

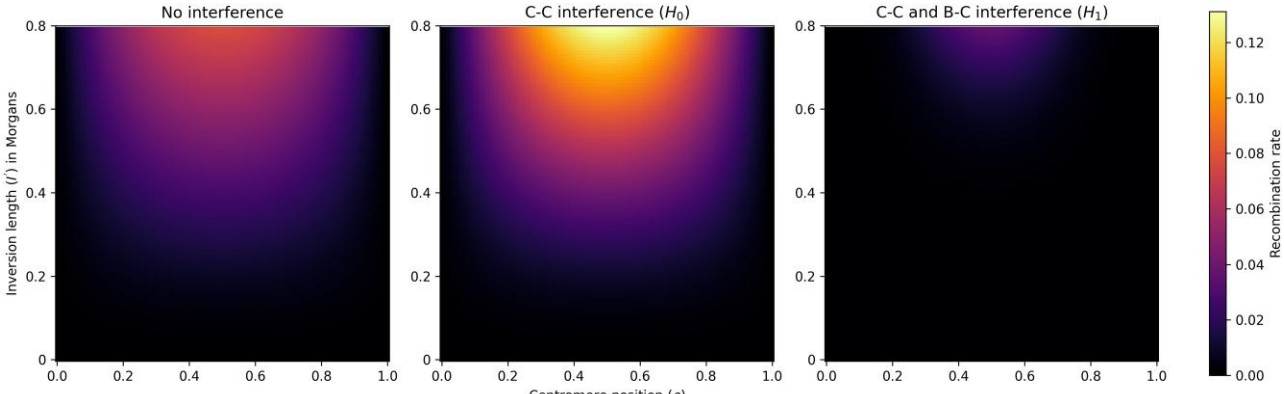

**Fig. 2.** The rate of recombination of three loci, of which two are linked to the breakpoints and one is linked to the centromere, as a function of the inversion length (I') and centromere position ($\rho$). The leftmost plot shows the case of no interference (independently distributed chiasmata), the middle plot shows the case of only chiasma–chiasma (C–C) interference (the $H_0$ model), and the rightmost plot shows the case of both chiasma–chiasma and breakpoint–chiasma (B–C) interference (the $H_1$ model). For the latter two, the strength of interference ($q_c$ and $q_c$, $q_b$, respectively) is set to 4, which is typical for *Drosophila* (Foss *et al.* 1993; McPeek and Speed 1995; Zhao *et al.* 1995; Lange *et al.* 1997; Kapperud 2023). The plots show that the recombination rate is increased when going from a state of no interference to a state of only chiasma–chiasma interference, but significantly decreased when going to a state of both chiasma–chiasma and breakpoint–chiasma interference.

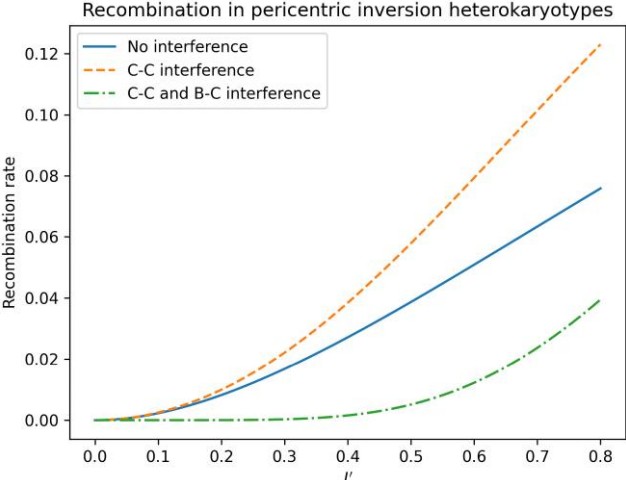

**Fig. 3.** For the same scenario as in Fig. 2, this figure shows a comparison of the recombination rates as a function of the inversion length (I') when $\rho = 0.5$.

the recombination rate in this scenario is increased when going from a state of no interference (independently distributed chiasmata) to a state of only chiasma interference (the $H_0$ model), but significantly decreased when also including breakpoint–chiasma interference (the $H_1$ model). Figures 4 and 5 similarly show that the sterility is highest when including only chiasma interference ($H_0$), at a middle level without any interference, and lowest when including both types of interference ($H_1$). I will return to these results in the *Discussion* section of this paper.

## Datasets and methods

I will consider three *Drosophila* pericentric inversion heterokaryotype recombination datasets for inversions *In(3LR)165*, *In(3LR)190* and *In(3LR)265* (see Fig. 6) published in Roberts (1967) and reproduced here in Supplementary Tables S1–S3. Like in Kapperud (2023), I will use the Nelder-Mead method to find the parameter values for $q_b$, $q_c$ and $l'_k$ (for each interval k) that maximizes the

likelihood. Comparing the estimated $l'_k$ values of the heterokaryotype datasets with the estimated $l_k$ values of a homokaryotype dataset (also from Roberts 1967), gives $d_k$ for all intervals k. $H_0$, $H_1$ and $H_2$ are not all nested, but they can be compared using the Akaike Information Criterium (AIC). $H_1$ and $H_2$ are nested, and can be compared using a likelihood ratio test, which tests the null hypothesis that the breakpoint–chiasma and chiasma–chiasma interferences are the same. The P values in the likelihood ratio tests are found by using the chi-square approximation. I will also use parametric bootstrap to find 95% confidence intervals for the estimated genetic lengths and interference strengths. Briefly, for each dataset and model, I first find the optimal parameter values using the Nelder-Mead method. I then create 99 resamples from the distribution found in the previous step, and perform the same analysis on each of these samples. The 95% confidence intervals are then found by discarding the most extreme 5% of values of each parameter over all analyses. For each sample, I repeat the Nelder-Mead method 100 times with different random starting points, so as to ensure that I find the global optima.

I will also consider the sterilities of the 30 inversions on chromosome 3 of *D. melanogaster* measured experimentally by Coyne *et al.* (1993). For these, I will use the Nelder-Mead method to find the optimal values of the strength of interference ($q_b$, $q_c$) and the overall degree of chiasma suppression that is not due to interference ($d_l$). The optimal values are in this case the ones that minimize the sum of the squared deviations between the observed and expected sterilities over all inversions. Note that to avoid overfitting, this analysis assumes that all the inversions in the dataset exhibit the same degree of chiasma suppression and interference, i.e. $d_l$ and $q_b$, $q_c$ are assumed to be the same for all inversions.

## Results

As Tables 2, 3 and 4 and Fig. 7 show, the breakpoint–chiasma and chiasma–chiasma interference model ($H_1$) performs markedly better than the model with only chiasma–chiasma interference ($H_0$) for all three pericentric inversions. Allowing the breakpoint–chiasma ($q_b$) and chiasma–chiasma ($q_c$) interference to vary independently ($H_2$) does not improve the likelihood by much, and the $H_1$ model is accordingly favored by the AIC for all three datasets. The likelihood ratio tests (Table 5) also indicate that we cannot

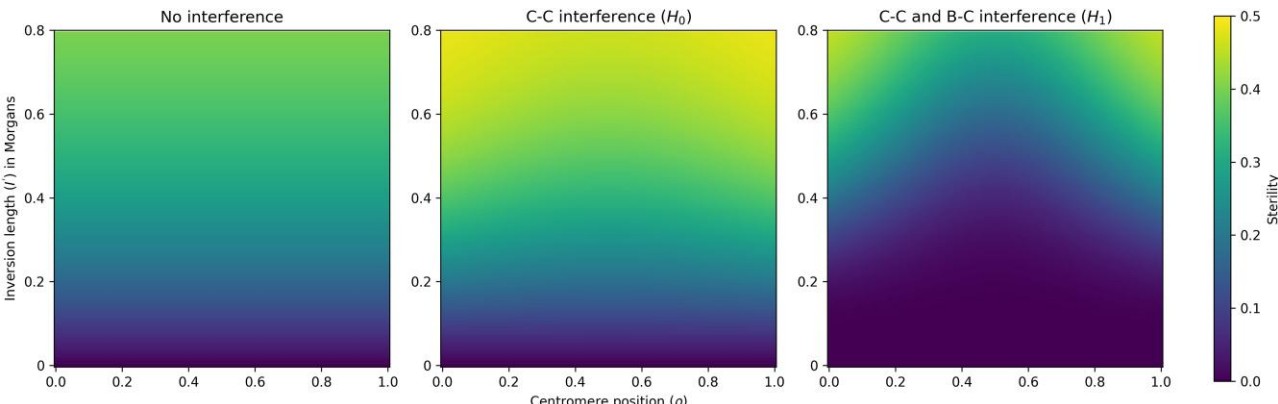

**Fig. 4.** The sterility of a pericentric inversion with no interference (independently distributed chiasmata), with chiasma–chiasma (C–C) interference (the $H_0$ model) and with chiasma–chiasma and breakpoint–chiasma (B–C) interference (the $H_1$ model). For the latter two, the strength of interference ($q_c$ and $q_c$, $q_b$, respectively) is set to 4, which is typical for *Drosophila* (Foss *et al.* 1993; McPeek and Speed 1995; Zhao *et al.* 1995; Lange *et al.* 1997; Kapperud 2023). The plots show that the probability of generating inviable gametes is increased when going from a state of no interference to a state of only chiasma–chiasma interference, but significantly decreased when going to a state of both chiasma–chiasma and breakpoint–chiasma interference.

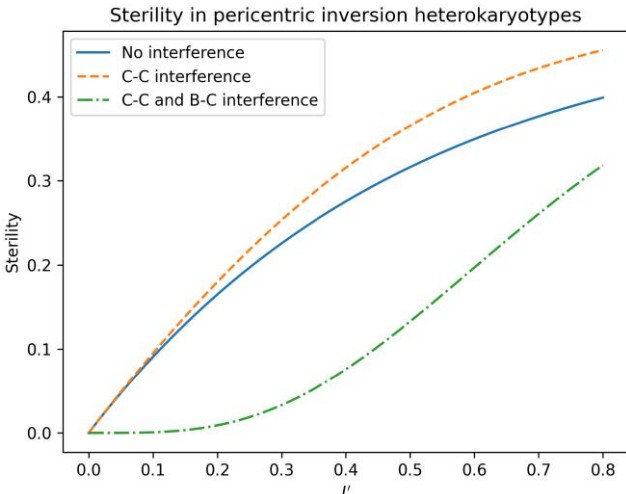

**Fig. 5.** For the same scenario as in Fig. 4, this figure shows a comparison of the sterilities as a function of the inversion length ($I'$) when $\rho = 0.5$.

reject the hypothesis that the two types of interference are the same. It appears, therefore, that the $H_1$ hypothesis (equal interference) is favored over both $H_0$ (no breakpoint interference) and $H_2$ (unequal interference).

Table 6 shows the estimated genetic maps of the control homokaryotype dataset and the three inversion datasets under the $H_1$ model. The maps mostly correspond fairly well ($d_k$ is close to 1 for most intervals), indicating that most, but not all, of the chiasma suppression is due to breakpoint interference rather than imperfect synapsis, in accordance with Gong *et al.*'s (2005) findings. The most notable deviation is that interval th-cu shows a substantial *increase* in genetic length for inversions 165 and 269 ($d_k \approx 5.4$ and $d_k \approx 8.3$, respectively). These are the only two cases where a breakpoint is located in the same interval as the centromere, and this finding therefore supports Koury's (2023) hypothesis that breakpoints, like chiasmata, interfere negatively in the pericentromeric region. In contrast, intervals that contain a centromere but not a breakpoint (th-cu for inversion 190) or a breakpoint but not a centromere (ve-h for inversion 165, ro-ca for inversion 269, h-th and cu-bx for inversion 190) do not show a comparable increase in genetic length.

Supplementary Table S4 lists all the chromosome 3 inversions investigated by Coyne *et al.* (1993). $I$ indicate the genetic length of the inversion in homokaryotypes and $\rho$ indicate the relative position of the centromere inside the inversion (on the standard genetic map). These are found by consulting Lindsley and Zimm (1992). The observed sterilities are the ones experimentally determined by Coyne *et al.* (1993). The expected sterilities for the $H_0$, $H_1$ and $H_2$ models are found by using the Nelder-Mead method to minimize the squared deviations between the expected and observed sterilities summed over all inversions, for the parameters $q_c$ and $q_b$ (strength of interference) and $d_l$ (degree of chiasma suppression that is not due to interference). As before, for $H_1$ $q_c = q_b$, whereas for $H_2$ they are allowed to vary independently. Table 7 shows the optimal values of $q_c$, $q_b$ and $d_l$ and the least squares (LS) for all three models. The LS for model $H_1$ is markedly lower than for $H_0$, indicating a preference for the former. The LS for $H_1$ and $H_2$ are the same, indicating no further improvement when allowing $q_c$ and $q_b$ to vary independently. (Repeated runs indicate that the $H_2$ model, unlike the other two, has multiple global optima, all of which with LS equal to that for the $H_1$ model. This can also be deduced directly from equations (9) and (10).) Additionally, the optimal strength of interference ($q_b = q_c = 3.72$) for the $H_1$ model is typical of chiasma interference in *Drosophila* (*e.g.* Foss *et al.* 1993; McPeek and Speed 1995; Zhao *et al.* 1995; Lange *et al.* 1997; Kapperud 2023) and the relatively high value of $d_l = 0.78$ is in line with Gong *et al.*'s (2005) observation of a pairing and synapsis frequency of 0.7-0.8 in a heterokaryotype. By contrast, the $H_0$ model suggests unrealistically strong interference ($q_c = 30$, the highest interference considered) and a much lower value of $d_l = 0.25$ (The latter is close to Navarro and Ruiz's (1997) estimate of $d_l = 0.27$ for the same data set, using a model without chiasma–chiasma or breakpoint–chiasma interference.) In sum, the analysis of the sterility data, like the analysis of the recombination data, supports the hypothesis that breakpoint–chiasma and chiasma–chiasma interference are the same.

## Discussion

I have in this paper shown how to incorporate breakpoint–chiasma interference into my earlier interference model, and used this extended model to analyse recombination and sterility data for *Drosophila* pericentric inversion heterokaryotypes. The results

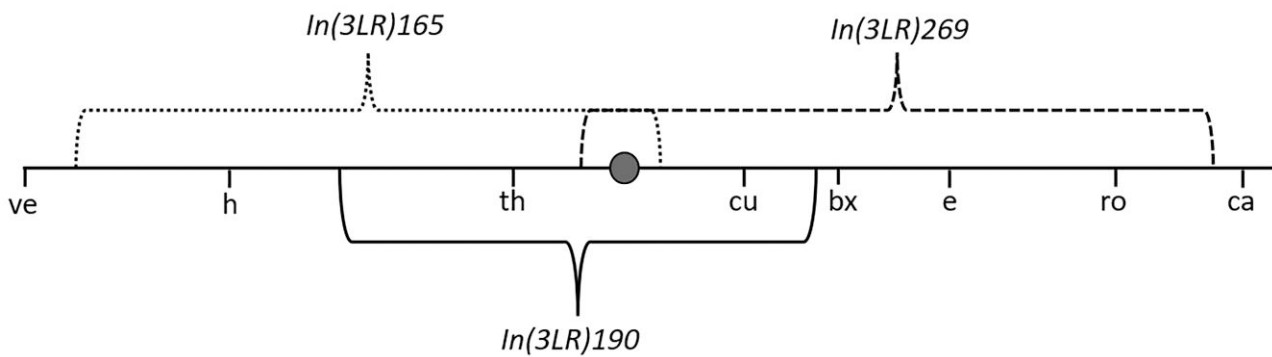

**Fig. 6.** The positions of inversions *In(3LR)165*, *In(3LR)190*, and *In(3LR)269* on chromosome 3 of *Drosophila melanogaster*, relative to the markers included in the recombination datasets and the centromere (gray circle).

**Table 2.** Optimal values of $q_c$ and $q_b$ (with 95% confidence intervals) and likelihood and AIC scores for inversion 190.

| Model | Parameters[a] | $q_c$ | $q_b$ | ln L | ΔAIC |
|---|---|---|---|---|---|
| $H_0$ | 8 | 6.81 (4.77–9.85) | – | −4,186.41 | 24.0 |
| $H_1$ | 8 | 8.57 (6.92–10.15) | 8.57 (6.92–10.15) | −4,174.41 | 0 |
| $H_2$ | 9 | 10.70 (5.63–12.43) | 15.00 (3.00–16.00) | −4,173.92 | 1.02 |

[a]Including 7 parameters for the genetic lengths of the 7 intervals.

**Table 3.** Optimal values of $q_c$ and $q_b$ (with 95% confidence intervals) and likelihood and AIC scores for inversion 269.

| Model | Parameters | $q_c$ | $q_b$ | ln L | ΔAIC |
|---|---|---|---|---|---|
| $H_0$ | 8 | 3.62 (2.65–4.74) | – | −6,813.74 | 7.30 |
| $H_1$ | 8 | 3.92 (2.95–4.91) | 3.92 (2.95–4.91) | −6,810.09 | 0 |
| $H_2$ | 9 | 3.49 (2.11–4.65) | 2.94 (0.91–5.86) | −6,809.83 | 1.48 |

**Table 4.** Optimal values of $q_c$ and $q_b$ (with 95% confidence intervals) and likelihood and AIC scores for inversion 165.

| Model | Parameters | $q_c$ | $q_b$ | ln L | ΔAIC |
|---|---|---|---|---|---|
| $H_0$ | 8 | 2.71 (1.98–3.68) | – | −5,402.00 | 3.34 |
| $H_1$ | 8 | 2.92 (2.53–3.77) | 2.92 (2.53–3.77) | −5,400.33 | 0 |
| $H_2$ | 9 | 3.06 (1.95–3.64) | 3.28 (1.17–5.03) | −5,400.22 | 1.78 |

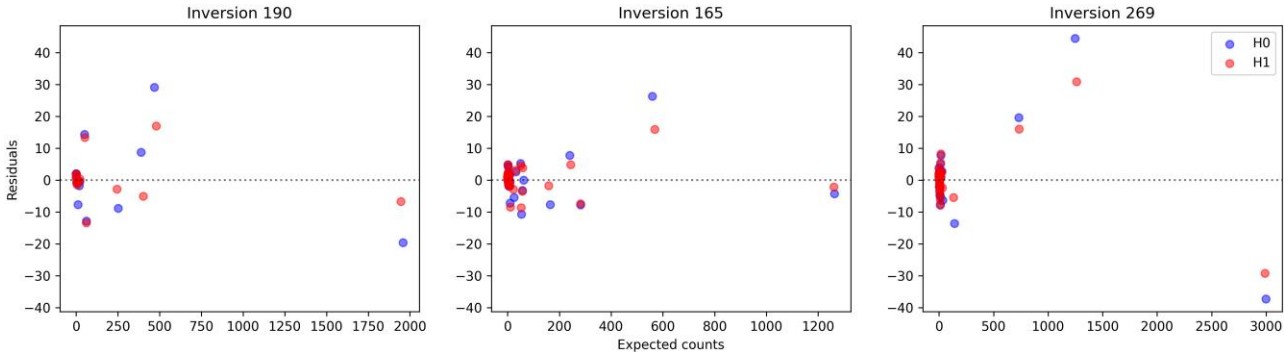

**Fig. 7.** Expected recombination pattern counts and residuals (observed minus expected counts) for inversions 190, 165, and 269, under the $H_0$ and $H_1$ models. Expected and observed counts for all three models and all three inversions are found in Supplementary Tables S1, S2, and S3.

**Table 5.** Likelihood ratio tests.

| Inversion | $H_1$ | $H_2$ | $\chi^2_1$ | P |
|---|---|---|---|---|
| 165 | −5,400.33 | −5,400.22 | 0.22 | 0.64 |
| 269 | −6,810.09 | −6,809.83 | 0.52 | 0.47 |
| 190 | −4,174.41 | −4,173.92 | 0.98 | 0.32 |

support Gong *et al.*'s (2005) hypothesis that inversion breakpoints interfere with chiasma formation in the same way and to the same degree that other chiasmata do. I also found a substantial increase in the genetic lengths of intervals containing both the centromere and an inversion breakpoint, which supports Koury's (2023) idea that breakpoints, like chiasmata (Green 1975;

**Table 6.** Genetic maps for the $H_1$ model compared to the homokaryotype control dataset.

| Inversion | ve-h | h-th | th-@-cu | cu-bx | bx-e | e-ro | ro-ca |
|---|---|---|---|---|---|---|---|
| *Control* | 27.6 cM (26.2–29.4) | 18.4 cM (17.1–19.6) | 4.5 cM (3.8–5.2) | 9.0 cM (8.0–9.7) | 15.8 cM (14.9–16.9) | 23.0 cM (21.5–24.5) | 11.4 cM (10.4–12.6) |
| 165 | <u>25.4 cM</u> (19.6–33.3) | 19.8 cM (10.7–29.7) | **<u>24.5 cM</u>** (20.3–40.3) | 15.5 cM (14.0–17.3) | 15.4 cM (14.0–17.0) | 28.0 cM (26.2–30.2) | 14.0 cM (13.1–15.0) |
| 269 | 27.1 cM (25.8–29.1) | 24.5 cM (22.8–25.9) | **<u>37.3 cM</u>** (23.7–41.0) | 16.2 cM (12.4–19.8) | 11.8 cM (9.4–14.5) | 12.1 cM (10.1–15.6) | <u>17.6 cM</u> (12.7–21.6) |
| 190 | 16.2 cM (14.4–17.9) | <u>21.4 cM</u> (18.5–24.0) | 1.0 cM (0.0–1.9) | <u>0.7 cM</u> (0.06–1.7) | 21.5 cM (18.9–23.2) | 21.0 cM (19.2–22.7) | 11.3 cM (10.0–12.4) |

Intervals containing an inversion breakpoint are underlined (ve-h and th-cu for inversion 165, th-cu and ro-ca for inversion 269, and h-th and cu-bx for inversion 190). The @ in the th-cu interval indicates the presence of the centromere in this interval. Intervals containing both an inversion breakpoint and the centromere are underlined in bold. 95% confidence intervals are shown beneath each table cell.

**Table 7.** Sterility statistics.

| Model | $q_c$ | $q_b$ | $d_l$ | LS |
|---|---|---|---|---|
| $H_0$ | 30.0 | – | 0.25 | 0.149 |
| $H_1$ | 3.72 | 3.72 | 0.78 | 0.0956 |
| $H_2$ | 3.72 | 3.72 | 0.78 | 0.0956 |

Sinclair 1975; Denell and Keppy 1979), interfere negatively in the pericentromeric region.

Although these results indicate that chiasmata and breakpoints may share some properties that are relevant for nearby chiasma distribution, they do not tell us what those properties are. Gong *et al.* (2005) and Koury (2023) suggest that the important factor is the similarities in the twisting of the chromosomal axis, but they leave unexplained why and how this twisting is supposed to cause interference in the first place. The answer to this conundrum may lie in recent findings suggesting that interference is due to the diffusion of crossover-promoting molecules through the synaptonemal complex, at least in *Arabidopsis* (Morgan *et al.* 2021; Fozard *et al.* 2023). In particular, HEI10 is hypothesized to cluster at the sites of double-strand breaks, and then diffuse in a coarsening process so that free HEI10 molecules are preferentially attracted to sites where many of them are already present. Hence, some sites will eventually attract all the HEI10 in their vicinity, leaving neighboring sites empty. If crossovers then only occur at sites where HEI10 is present, it would explain why chiasmata appear to interfere with the generation of nearby chiasmata. For this discussion, the relevant question then becomes what happens with the HEI10 molecules at inversion breakpoints. The answer is not currently known (Morgan *et al.* 2024), but Morgan *et al.* (2024) speculate that impeded or altered HEI10 diffusion may be the key to explaining the absence of chiasmata close to breakpoints. For example, breakpoints could serve as HEI10 sinks (attracting nearby HEI10 molecules) or leaks (allowing HEI10 molecules to leak out into the nucleoplasm) or a combination of the two. Such processes would remove HEI10 from the vicinity of the breakpoints, and thus inhibit the formation of nearby chiasmata, similarly to how chiasma-to-be double-strand breaks do the same. The finding in this paper that chiasma–chiasma and breakpoint–chiasma interference are the same in magnitude may indicate that chiasmata and breakpoints attract HEI10 molecules in a similar way or at least at a similar rate. This idea could be tested further by extending the existing mathematical models of the HEI10 coarsening process (Morgan *et al.* 2021; Fozard *et al.* 2023) to account for different hypothesized effects of the breakpoints on HEI10 diffusion, and comparing the outcome to real-life data. This approach would have the advantage of being applicable for both pericentric and paracentric inversions, as it would solve the problem raised by Koury (2023) of how to model the interaction between the interferences from the two breakpoints in the latter case. Immunolocalization could also be used to investigate the distribution of HEI10 molecules during meiosis in inversion heterokaryotypes, similarly to what has been done in homokaryotypes (Morgan *et al.* 2021; Fozard *et al.* 2023).

The HEI10 coarsening hypothesis is well supported in *Arabidopsis* (Morgan *et al.* 2021; Fozard *et al.* 2023) and a similar model has been proposed for *C. elegans* (Zhang *et al.* 2021), but to my knowledge it is yet to be tested in *Drosophila*. The known meiotic differences between different groups of organisms (e.g. Stahl *et al.* 2004; Kapperud 2023) therefore call for some caution in interpreting the current results in light of this hypothesis. If future investigations reveal that a similar coarsening process takes place in all eukaryotes, it could present a biological interpretation of the elusive "machine that can count" sought by the early counting model pioneers (Foss *et al.* 1993; Foss and Stahl 1995). That is, in this interpretation the independently distributed dummy events in the counting models are non-chiasmatic double-strand breaks and the HEI10 coarsening is the process that causes such events to reliably intervene between the chiasma events. The hypothesis does not, however, explain why chiasma interference exists in the first place. The prevailing answer to this question is that chiasma interference evolved as a way of ensuring that at least one crossover event occurs on each chromosome (*e.g.* Otto and Payseur 2019), an idea called *crossover assurance*. In this view, breakpoint–chiasma interference exists as a byproduct of chiasma interference. However, breakpoint–chiasma interference is also adaptive in its own right, since discouraging chiasma formation close to inversion breakpoints significantly lowers the sterility effect of the inversion, as shown in Figs. 4 and 5. It is also known that there is selection for restricting recombination between sets of genes that are locally adapted or show favorable epistatic interactions (Kirkpatrick and Barton 2006), and breakpoint–chiasma interference would therefore introduce the double advantage of limiting sterility while reducing recombination between such genes inside of or close to heterozygous inversions (Figs. 2 and 3), thus allowing the inversions to spread in the population. Accordingly, locally adapted genes are often found inside inversions (e.g. Sinclair-Waters *et al.* 2018; Koch *et al.* 2021). This suggests the possibility that breakpoint–chiasma interference evolved as a response to the selection pressure introduced by chromosomal inversions, and that chiasma interference came along as a byproduct, rather than the other way around. The two phenomena could also have evolved sequentially, with breakpoint interference evolving first, and chiasma interference evolving later by exapting the existing molecular machinery, or vice versa. It is also possible, however, that interference is an inherent property of meiotic recombination, and that there never was a state of

no-interfering recombination from which interference could evolve. This would make both chiasma and breakpoint–chiasma interference a *spandrel* in Gould and Lewontin's (1979) terminology.

The relative merit of these hypotheses can be evaluated through modeling or comparative genetics. For example, the hypothesis that chiasma interference evolved first assumes that the adaptive advantage of crossover assurance is larger than the disadvantage of increasing sterility and recombination in inversion heterokaryotypes, absent breakpoint interference (Figs. 2, 3, 4, and 5). Simulations and mathematical models can tell us under which circumstances this assumption is reasonable. Similarly, the hypothesis that breakpoint interference evolved first assumes that there would first be selection for an intermediate stage of breakpoint–chiasma interference without chiasma–chiasma interference, and then further selection for a stage with both types of interference. These assumptions could also be investigated through modeling. And finally, comparing the molecular mechanisms of interference in different species, including those that lack interference altogether, may help answering the question of whether the two types of interference evolved sequentially or simultaneously, or whether they were likely both present when recombination first arose.

## Data availability

I have only analyzed previously published datasets from Roberts (1967) and Coyne *et al.* (1993). These are also reproduced in Supplementary Tables S1–S4. The code used to perform the analysis is available at https://doi.org/10.6084/m9.figshare.28716044 and github.com/kpprd/recombination. The former is a static version, whereas the latter may be updated with extended documentation and additional features for future projects.

Supplemental material available at GENETICS online.

## Acknowledgments

I thank three anonymous reviewers for comments and helpful suggestions.

## Funding

I have not received funding for this work.

## Conflicts of interest

The author(s) declare no conflicts of interest.

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

*Editor: W. Valdar*