## [Peer Review File · Genetics]

Breakpoint-chiasma interference in pericentric inversion heterokaryotypes

Øystein Kapperud

NOTE: The reviews and decision letters are unedited and appear as submitted by the reviewers.

In extremely rare instances and as determined by a Senior Editor or the EIC, portions of a review may be redacted. If a review is signed, the reviewer has agreed to no longer remain anonymous.

The review history appears in chronological order.

Review Timeline:

Submission Date:	2024-08-04
Editorial Decision:	2024-10-20
Resubmission Received:	2025-03-03
Accepted:	2025-03-31

October 20, 2024

GENETICS-2024-307347

Breakpoint-chiasma interference in pericentric inversion heterokaryotypes

Dear Dr. Kapperud:

Three experts in the field have reviewed your manuscript, and I have read it as well. All reviewers felt that the work described in the manuscript provided a potentially useful improvement to the existing model. However, one reviewer felt that the elaborations to the model needed more justification and would benefit from consideration of ideas from additional recent literature on coarsening. There were also mixed opinions on whether the extent of the improvement makes the work of sufficiently broad interest for GENETICS. Additionally, the provision of software and data for the purposes of reproducibility fell short of requirements.

Although your manuscript is not currently acceptable for publication in GENETICS, we would welcome a substantially revised manuscript. All three reviewers have comments and concerns to be addressed in a revised manuscript. You can read their reviews at the end of this email.

In a revised manuscript, I will expect to see a point-by-point response to all the questions posed by the reviewers. In particular:

1. You must provide the software and data files you used as per GENETICS author instructions. The entire analysis should be repeatable by a reader.
2. Address Reviewer 3's comments about coarsening models. It sounds like this should be discussed and potentially addressed not only in your response but also in the manuscript itself.
3. Include diagnostic plots and contingency tables as per Reviewer 2's suggestions, either in the main manuscript or in Supplemental Materials.

We look forward to receiving your revised manuscript. Please let the editorial office know approximately how long you expect to need for revisions.

Upon resubmission, please include:

1. A clean version of your manuscript;
2. A marked version of your manuscript in which you highlight significant revisions carried out in response to the major points raised by the editor/reviewers (track changes is acceptable if preferred);
3. A detailed response to the editor's/reviewers' feedback and to the concerns listed above. Please reference line numbers in this response to aid the editor and reviewers.

Your paper will likely be sent back out for review.

Additionally, please ensure that your resubmission is formatted for GENETICS
<https://academic.oup.com/genetics/pages/general-instructions>

Follow this link to submit the revised manuscript: Link Not Available

Sincerely,

William Valdar
Associate Editor
GENETICS

Approved by:
Hongyu Zhao
Senior Editor
GENETICS

Reviewer #1 :

After almost a hundred years of modelling recombination within inversions and its effects; after a huge amount of literature on the matter, it is very pleasing to see the second paper of an author that, almost single-handedly, is pushing the field forward.

On this occasion, the paper addresses a longstanding and complex biological problem-chiasma interference, particularly in the context of pericentric inversion heterokaryotypes in *Drosophila melanogaster*. The author provides a novel and detailed model to explain interference between inversion breakpoints and chiasmata. The study uses an impressive collection of data, as well as sophisticated modeling techniques, to support the hypothesis that breakpoints interfere with chiasma formation similarly to other chiasmata. The manuscript is well-structured and offers significant contributions to our understanding of meiotic recombination in inversion heterokaryotypes.

In summary, this manuscript addresses a critical biological question with a well-supported and innovative approach. It provides valuable insights into the mechanisms governing chiasma formation in inversion heterokaryotypes and offers exciting directions for future research. I have no major comments. At most, I have some suggestions that the author may choose not to follow, and a few minor comments on typos and wording.

SUGGESTIONS

- Theoretical Framework. The paper presents a new theoretical model and compares it to the better known ones. Adding more explanation on how these models differ in terms of biological significance, rather than just statistical outcomes, could enhance understanding for a broader audience.

- Implications and future directions. In the last paragraph of the paper, the author brilliantly suggests that breakpoint-chiasma interference may be either adaptive or a spandrel, at any rate lowering sterility effects in inversion heterokaryotypes. While this is an intriguing idea, it could be helpful to expand on potential evolutionary implications. A more detailed discussion of how these two hypotheses can be "disentangled" (to use his word) and how they might affect inversion frequency and survival in natural populations would be an interesting addition to the Discussion section. Providing a more concrete outline of potential future models, simulations, experiments or analyses that could be conducted to carry out that test would be great. .

- Figures:: Figure 2 could benefit from revisiting. Perhaps a more detailed legend explaining the different panels and what they represent for readers less familiar with field.

MINOR COMMENTS:

Typographical Errors:

- On page 2, line 25: "mulitplication" should be corrected to "multiplication."
- On page 3, line 42: "realtive" should be corrected to "relative."

Reviewer #2 :

The author expands on their recent paper on a general model for chiasma interference by considering a model in which inversion breakpoints may similarly interfere with nearby chiasmata. They apply their approach to three *Drosophila* datasets from Coyne et al. (1993). In the analysis, they compare three models:

H0 that interference is blocked by the centromere and inversions, but that inversions do not interfere with nearby chiasmata

H1 that interference is blocked by the centromere and that inversions act much like chiasmata in regard to interference

H2 like H1 but where the inversion:chiasma interference may be different from the chiasma:chiasma interference

In each of the three datasets, model H1 shows the best fit.

1. The analysis appears solid. It would be nice if there were some other diagnostic plots that might be informative about H0 vs H1, such as

a scatterplot or 2-way contingency table of recombination data.
The likelihood calculations don't provide much intuitive insight.

2. The author should provide the software they used for the analysis.

3. I think Table 7 might be better shown as a multi-panel figure, or perhaps included as a supplement. I don't quite understand what we learn from this table.

Reviewer #3 :

This paper is a follow-up from a publication from this same author last year, in which he modeled the effects of inversion breakpoints on crossover distribution if inversion breakpoints block a crossover interference signal. Inspired by Koury (2023), he now presents a revised model in which the breakpoints themselves exert interference. This model appears to fit existing data well, and is therefore an advance in understanding meiotic chromosomes and the effects of chromosome rearrangements.

In this reviewer's opinion, the justification for the model is not strong. I'm not aware of any evidence that rearrangement breakpoints actually resemble chiasmata. Gong et al. offered this speculation to explain the effects seen in balancer heterozygotes. EM of recombination nodules, which are perhaps the highest resolution images of crossover sites, sit on top of the SC and, to my knowledge, do not alter the structure of the SC below (indeed, in Carpenter's initial report of the discover of RNs she noted that in a couple cases they were actually SC-adjacent). In this view, there is no a priori reason to expect rearrangement breakpoints to exert interference.

I believe this modeling would benefit by incorporating ideas from a recent model of interference. Morgan et al. (doi 10.1038/s41467-021-24827-w) and Zhang et al. (doi 10.1101/2021.08.26.457865) suggested that crossover designation occurs through the biophysical process of coarsening, in which a protein diffusing through the SC tends to accumulate more and more until it is all at one site (for single crossover); when there is more than one site (multiple crossover bivalents) they must be far apart, or they would exchange subunits until one of the sites has them all. In this model, any SC that has a DSB and enough of this molecule to designate a crossover will have (at least) one crossover. This is crossover assurance; interference is merely a byproduct of this process.

This hypothesis suggests yet another possibility for the effects of rearrangement breakpoints: They may function like ends of the SC. If most of the source of the crossover designation molecules is diffusion within individual SCs (with perhaps some intranuclear diffusion between SCs), then there will be fewer crossovers near ends, where these molecules can come from only one direction. If a rearrangement breakpoint results in a discontinuity in the SC, at least with respect to diffusion, then that point will behave the same as the end of a chromosome. To this reviewer the idea that a change in pairing partners causes a discontinuity in the SC seems more likely than the idea that these sites behave like crossover sites and send out an interference signal.

While the current manuscript is a worthwhile addition to the literature as it stands, I feel it would be more impactful if it included the ideas suggested by the coarsening model. Given the rapid rise in popularity of the proposal that coarsening explains crossover designation and interference, I fear this manuscript in its present form may feel outdated more quickly than it should.

Associate Editor Comments:

Dear editors and reviewers

This text is a response to the feedback I received on the manuscript *Breakpoint-chiasma interference in pericentric inversion heterokaryotypes* that I submitted to Genetics last autumn. I am grateful for the helpful comments and suggestions, and have now revised the manuscript accordingly. The most notable changes are that I have included a longer discussion about the biological significance and evolutionary implications of the results, as suggested by reviewer 1, a discussion of the HEI10 coarsening model, as requested by reviewer 3, and more plots and tables, as requested by reviewer 2. I have also indicated where the reader can find the code used to perform the analysis (in the *Data availability* section). In the remainder of this text, I will respond to the reviewers' feedback in more detail and indicate where in the text the changes have been made. Major changes are also highlighted in the accompanying file *manuscript_changes.pdf*. The reviewers' comments are in italics and my responses are roman in the following.

Reviewer #1 :

After almost a hundred years of modelling recombination within inversions and its effects; after a huge amount of literature on the matter, it is very pleasing to see the second paper of an author that, almost single-handedly, is pushing the field forward.

*On this occasion, the paper addresses a longstanding and complex biological problem-chiasma interference, particularly in the context of pericentric inversion heterokaryotypes in *Drosophila melanogaster*. The author provides a novel and detailed model to explain interference between inversion breakpoints and chiasmata. The study uses an impressive collection of data, as well as sophisticated modeling techniques, to support the hypothesis that breakpoints interfere with chiasma formation similarly to other chiasmata. The manuscript is well-structured and offers significant contributions to our understanding of meiotic recombination in inversion heterokaryotypes.*

In summary, this manuscript addresses a critical biological question with a well-supported and innovative approach. It provides valuable insights into the mechanisms governing chiasma formation in inversion heterokaryotypes and offers exciting directions for future research. I have no major comments. At most, I have some suggestions that the author may choose not to follow, and a few minor comments on typos and wording.

Thank you very much for the encouraging feedback!

SUGGESTIONS

- Theoretical Framework. The paper presents a new theoretical model and compares it to the better known ones. Adding more explanation on how these models differ in terms of biological significance, rather than just statistical outcomes, could enhance understanding for a broader audience.

In the revised manuscript, I have included a short discussion and plots (figures 3 and 4) of the effect of breakpoint interference on recombination, similar to the previous discussion of the effect on sterility. I have combined these discussions into a single paragraph and placed it at the end of the Models section (page 3, line 88 onwards). I have also included a discussion of the biological significance of the breakpoint-chiasma interference model in light of the HEI10 coarsening hypothesis (discussion section, page 6), as suggested by reviewer 2.

- Implications and future directions. In the last paragraph of the paper, the author brilliantly suggests that breakpoint-chiasma interference may be either adaptive or a spandrel, at any rate lowering sterility effects in inversion heterokaryotypes. While this is an intriguing idea, it could be helpful to expand on potential evolutionary implications. A more detailed discussion of how these two hypotheses can be "disentangled" (to use his word) and how they might affect inversion frequency and survival in natural populations would be an interesting addition to the Discussion section. Providing a more concrete outline of potential future models, simulations, experiments or analyses that could be conducted to carry out that test would be great.

I have expanded the Discussion section of the manuscript in accordance with these suggestions.

- Figures: Figure 2 could benefit from revisiting. Perhaps a more detailed legend explaining the different panels and what they represent for readers less familiar with field.

I have included a more non-technical interpretation of the plots and placed them at the end of the *Models* section, rather than in the middle, as was previously the case. This gives the reader more background before encountering the plots. I have also included additional plots (figures 4 and 6) that compare the recombination and sterility more directly, for a given position of the centromere, and a short explanation, with references, of counting models in general (page 1, line 40 onwards), to make the text more accessible to those who are not familiar with such models.

MINOR COMMENTS:

Typographical Errors:

- On page 2, line 25: "multiplication" should be corrected to "multiplication."

- On page 3, line 42: "realitive" should be corrected to "relative."

I have corrected the typos in the revised manuscript.

Reviewer #2 :

The author expands on their recent paper on a general model for chiasma interference by considering a model in which inversion breakpoints may similarly interfere with nearby chiasmata. (..) In each of the three datasets, model H1 shows the best fit.

1. The analysis appears solid. It would be nice if there were some other diagnostic plots that might be informative about H0 vs H1, such as a scatterplot or 2-way contingency table of recombination data. The likelihood calculations don't provide much intuitive insight.

I have now included a plot of the residuals in the main text and full tables of expected and observed counts in the supplemental material.

2. The author should provide the software they used for the analysis.

The code is available at github.com/kpprd/recombination. I have now also indicated this in the manuscript under *Data availability*.

3. I think Table 7 might be better shown as a multi-panel figure, or perhaps included as a supplement. I don't quite understand what we learn from this table.

I have now moved this table to the supplemental material. I think the table is necessary for allowing the reader to easily check and repeat the analysis, as it shows information not readily available elsewhere (I calculated the inversion lengths and centromere positions based on information available in Lindsley and Zimm 1992, but doing so was quite tedious).

Reviewer #3 :

This paper is a follow-up from a publication from this same author last year, in which he modeled the effects of inversion breakpoints on crossover distribution if inversion breakpoints block a crossover interference signal. Inspired by Koury (2023), he now presents a revised model in which the breakpoints themselves exert interference. This model appears to fit existing data well, and is therefore an advance in understanding meiotic chromosomes and the effects of chromosome rearrangements.

In this reviewer's opinion, the justification for the model is not strong. I'm not aware of any evidence that rearrangement breakpoints actually resemble chiasmata. Gong et al. offered this speculation to explain the effects seen in balancer heterozygotes. EM of recombination nodules, which are perhaps the highest resolution images of crossover sites, sit on top of the SC and, to my knowledge, do not alter the structure of the SC below (indeed, in Carpenter's initial report of the discover of RNs she noted that in a couple cases they were actually SC-adjacent). In this view, there is no a priori reason to expect rearrangement breakpoints to exert interference.

This is a fair point, but I do not think the validity of the model necessarily rests on whether or not breakpoints resemble chiasmata. The results I present in the manuscript suggest that breakpoint-chiasma and chiasma-chiasma interference are similar in two important respects: they are similar in magnitude and have the same effect (being negative rather than positive) in the centromeric region. This suggests to me that they share some properties relevant for interference. The language I used in the previous manuscript heavily implied that the shared property is the twisting of the chromosomal axis, but I agree with the reviewer that is not necessarily the case. I have therefore amended the introduction to make the language more neutral with regards to what the shared properties may be (see page 1, line 2; page 1, line 6 onwards), and included a longer discussion of this matter in the discussion section (page 6), in particular with regards to the HEI10 coarsening model, as suggested by the reviewer.

I believe this modeling would benefit by incorporating ideas from a recent model of interference. Morgan et al. (doi 10.1038/s41467-021-24827-w) and Zhang et al. (doi 10.1101/2021.08.26.457865) suggested that crossover designation occurs through the biophysical process of coarsening, in which a protein diffusing through the SC tends to accumulate more and more until it is all at one site (for single crossover); when there is more than one site (multiple crossover bivalents) they must be far apart, or they would exchange subunits until one of the sites has them all.

I agree with this point, and have included a longer discussion of the HEI10 coarsening model in the Discussion section.

In this model, any SC that has a DSB and enough of this molecule to designate a crossover will have (at least) one crossover. This is crossover assurance; interference is merely a byproduct of this process.

I acknowledge in the text that this the prevailing theory of why chiasma interference exists. It may very well be true, but I think it is worth raising and discussing alternative theories, especially since the crossover assurance theory seems almost universally accepted despite there being (to my knowledge) little direct evidence either way on whether it is true.

This hypothesis suggests yet another possibility for the effects of rearrangement breakpoints: They may function like ends of the SC. If most of the source of the crossover designation molecules is diffusion within individual SCs (with perhaps some intranuclear diffusion between SCs), then there will be fewer crossovers near ends, where these molecules can come from only one direction. If a rearrangement breakpoint results in a discontinuity in the SC, at least with respect to diffusion, then that point will behave the same as the end of a chromosome.

This is an interesting idea. In the text, I credit Morgan et al (2024) with a similar idea and expand on it with some of my own (page 6, line 19 onwards). I will say that whether or not the breakpoints behave like the end of the chromosome depends on the details of the impeded diffusion, in particular whether or not the molecules are completely removed from the SC or partly diffuse back in the other or both directions. In the text, I suggest that the similarities between breakpoint and chiasma interference may indicate that they attract HEI10 molecules at a similar rate, assuming a similar coarsening process also occurs in *Drosophila*. I also suggest some other ways of testing this prediction in future research projects.

To this reviewer the idea that a change in pairing partners causes a discontinuity in the SC seems more likely than the idea that these sites behave like crossover sites and send out an interference signal.

I do not think these are necessarily incompatible. If the interference signal is caused by the diffusion of HEI10, then impediments to this diffusion could be equivalent to sending out an interference signal. However, I see now that this language and the language I used in the original manuscript ("sending out an interference signal", "the signal propagates in both directions", etc) may be misleading, as it seems to imply that the interference happens linearly and propagates in a given direction, which is not necessarily the case. The counting models do not rest on the interference happening in any particular way, and so is not incompatible with a coarsening process. The recombination patterns are calculated linearly one interval at the time, but this is merely a mathematical convenience, not a reflection on how the interference actually propagates. I have amended the text to make this point clearer. See in particular page 3, line 5 onwards.

While the current manuscript is a worthwhile addition to the literature as it stands, I feel it would be more impactful if it included the ideas suggested by the coarsening model. Given the rapid rise in popularity of the proposal that coarsening explains crossover designation and interference, I fear this manuscript in its present form may feel outdated more quickly than it should.

I agree with this point, as discussed above.

One additional noteworthy change is that I have made it explicit that the genetic maps for the H1 model indicate that most of the chiasma suppression is due to breakpoint interference (page 4, line 58 onwards). This was only implicit in the previous version of the manuscript.

I hope you will find that the changes improve the manuscript sufficiently to be acceptable for publication, but please let me know if you have suggestions for further improvements. I wish you a good day and look forward to hearing from you again.

Øystein Kapperud

March 31, 2025

RE: GENETICS-2025-307944

Dr. Øystein Kapperud
Folkehelseinstituttet
IT Systems Oslo
Norwegian Institute of Public Health, P.O Box 222 Skøyen
Oslo, N/A 0213
Norway

Dear Dr. Kapperud:

Congratulations! We are delighted to inform you that your manuscript titled "Breakpoint-chiasma interference in pericentric inversion heterokaryotypes" is acceptable for publication in GENETICS. Many thanks for submitting your research to the journal.

Regarding your data availability, your github link fulfills the requirement of being publically accessible. Nonetheless, I encourage you to make available a "frozen" version of the code on a static repository like figshare, since github repos can change substantially over time.

To Proceed to Production:

1. Format your article according to GENETICS style, as discussed at <https://academic.oup.com/genetics/pages/general-instructions>, and upload your final files at <https://genetics.msubmit.net>.
2. Your manuscript will be published as-is (unedited-as submitted, reviewed, and accepted) at the GENETICS website as an Advanced Access article and deposited into PubMed shortly after receipt of source files and the completed license to publish. Please notify sourcefiles@thegsajournals.org if you do not wish to publish your article via Advanced Access.
3. We invite you to submit an original color figure related to your paper for consideration as cover art. Please email your submission to the editorial office or upload it with your final files. You can submit a small-sized image for evaluation, and if selected, the final image must be a TIFF file 2513px wide by 3263px high (8.375 by 10.875 inches; resolution of 600ppi). Please avoid graphs and small type.

If you have any questions or encounter any problems while uploading your accepted manuscript files, please email the editorial office at sourcefiles@thegsajournals.org.

Sincerely,

William Valdar
Associate Editor
GENETICS

Approved by:
Hongyu Zhao
Senior Editor
GENETICS

note: Please add jnls.author.support@oup.com and genetics.oup@kwglobal.com (or the domains @oup.com and @kwglobal.com) to your email program's "safe senders" list. You will be contacted by both at various points during the production process.

Review comments (if applicable):

Reviewer #1 :

Congratulations on a much better paper. Also, congratulations to you and my fellow reviewers on brilliant points in their reviews

and on a very well argued Covering Letter. I enjoyed the exchange. It is unusual that reviewing is so fun!

Reviewer #3 :

The author has addressed my comments. In particular, the new discussion of implications for the coarsening model is thoughtful and thought-provoking.